# Numerical Simulation of Non-Isothermal Mixing Flow Characteristics with ELES Method

Chengbin Sun [1,*], Hexu Wang [2], Yanlong Jiang [1], Zhixin Zou [3] and Faxing Zhu [1]

[1] Key Laboratory of Aircraft Environment Control and Life Support, MIIT, Nanjing University of Aeronautics and Astronautics, 29 Yudao St., Nanjing 210016, China; jiang-yanlong@nuaa.edu.cn (Y.J.); faxing_zhu@nuaa.edu.cn (F.Z.)

[2] Aviation Key Laboratory of Science and Technology on Aero Electromechanical System Integration, Nanjing 211106, China; wdj20120526@163.com

[3] China National Aeronautical Radio Electronics Research Institute, Shanghai 200241, China; 17551066653@163.com

* Correspondence: suncb@nuaa.edu.cn

**Abstract:** Thermal fatigue caused by turbulent thermal mixing in tee pipes is always one of the failure factors of industrial pipes. At present, computational fluid dynamics (CFD) is still the main research method to study the thermal fatigue mechanism. Due to the limitations of the large eddy simulation (LES) model and the classical Reynolds Averaged Navier–Stokes (RANS) model in simulating thermal mixing, an advanced Embedded LES (ELES) model was developed. By comparing the model with data in the open literature, the validity of the ELES model to iso-thermal mixing was evaluated and proven. After that, the flow characteristics of the backflow upstream with different momentum ratios ($M_R$) were studied using the ELES method, as well as the temperature characteristics near the wall where the backflow appears. It was found that the characteristics of the backflow and the temperature distribution upstream in the tee were different with different $M_R$ values and some regions under specified $M_R$ values are found to be more prone to thermal fatigue at the intersection of the tee upstream. Moreover, the frequency analysis at specified points near the wall under three different $M_R$ values was estimated to evaluate thermal fatigue and the results showed that long-period fluctuations of lower frequencies than 6 Hz upstream were observed. This work helps form a comprehensive understanding of the backflow in thermal mixing and the relationship between fatigue and backflow in the tee.

**Keywords:** tee pipe; large eddy simulation; non-isothermal mixing; ELES; thermal fatigue

## 1. Introduction

Since several leakages caused by thermal striping in industrial tee pipes have occurred, notably the accidents that happened in the light water reactor of French PWR Civaux 1, 1998 [1], and in the sodium-cooled fast reactor of PHENIX, 1991 [2], they have aroused wide attention of scholars to focus on the thermal fatigue caused by turbulent thermal mixing. In order to study the relationship between temperature and crack, various experiments have been conducted for temperature fluctuation in the mixing tees. For example, Simoneau et al. [3] and Wakamatsu et al. [4] have conducted a scaled model experiment related to Civaux 1. Furthermore, water experiments (WATLON) have been conducted by Kamide et al. [5] in the Japan Atomic Energy Agency (JAEA) to study the mixing behavior in tee pipes using particle image velocimetry (PIV) and thermocouples. However, due to the limitations of the experimental conditions and the means of observation, computational fluid dynamics (CFD) has become a cheaper and more efficient method to study the flow characteristics of thermal mixing.

So far, many numerical methods have been used to study the turbulence characteristics of mixing flows. Undoubtedly, the Reynolds-Averaged Navier–Stokes (RANS) turbulence

models are the most classical and basic. Frank et al. [6], for example, employed RANS turbulence models to simulate the non-isothermal mixing flows in tee pipes using ANSYS CFX software. It was found that the RANS model can accurately simulate the turbulent mixing of isothermal fluids in tee pipes but failed in non-isothermal mixing flows. Additionally, throughout the past few decades, the large eddy simulation (LES) model has been the most used model to simulate free shear flow. For example, Hu and Kazimi [7] carried out a numerical study with LES to investigate the impact of the temperature difference between the mixing flows on temperatures downstream of the tee pipes. Odemark et al. [8] performed an LES analysis of turbulent mixing in a tee pipe with different flow ratios. Tanaka et al. [9] and Utanohara et al. [10] ran numerical simulations with LES approaches for WATLON of a mixing tee in JAEA to investigate the mechanism of thermal fatigue based on the work of Kamide et al. [5]. Other studies using LES were performed by Lee et al. [11], Jayaraju et al. [12], and Kuhn et al. [13] on different aspects of mixing flows. In recent years, several kinds of hybrid RANS-LES models such as Delayed Detached Eddy Simulation (DDES) and Scale-Adaptive Simulation (SAS) were developed for resolving turbulent structures. Zeng et al. [14], Krumbein et al. [15], and Chang et al. [16] employed different hybrid RANS-LES models to simulate the mixing flows in tee pipes. It was found that the results of hybrid RANS-LES models are very close to the experimental data and that of LES.

However, the RANS model, the LES model, and the two hybrid RANS-LES models (DDES and SAS) have their own limitations. Compared with the LES model, there are two main limitations of the RANS model. The first one is the lack of additional information obtained from the RANS simulation, especially for unsteady mixing flows with different temperatures. The second one is related to accuracy. According to the previous study, it is evident that the performance of LES models for free shear flows is significantly more consistent than that of RANS [17,18]. As for the LES model, the high computational cost required for LES at high or even at moderate Reynold numbers is unacceptable [19] because the turbulence scales in LES formulations are of the order of the shear layer thickness, which makes the turbulence length scale extremely small in relation to the thickness of the boundary layer close to the wall [20]. With regard to the hybrid RANS-LES models (DDES and SAS), they may not always be suitable, particularly for flows without strong enough instability to produce turbulent structures on their own [21]. In some conditions, the instability of the separating shear layer cannot be captured by the models, even though resolved turbulence is described at the entrance because the models typically switch back to RANS mode after some boundary layer thicknesses.

Nowadays, a new hybrid RANS-LES model is introduced, which is called the Embedded Large Eddy Simulation (ELES) approach. The ELES model has solved the above problem through the predefinition of interfaces in the geometric model. By this method, the ELES model can not only reduce the computational cost but can also obtain good predictive accuracy of the solution by distinguishing RANS and LES regions and creating synthetic turbulence at RANS-LES interfaces that have been predefined [22]. According to the literature [19], DDES, SAS, and ELES were compared, and it was revealed that better temperature predictions and stronger resolved turbulence activity were found by using the ELES approach than by using the two other hybrid models.

In order to fully understand the flow characteristics and fatigue mechanism in tee pipes, the non-isothermal mixing flow characteristics were investigated using the ELES method with commercial computational Fluid Dynamics (CFD) software ANSYS FLUENT 17.0 in this paper. Firstly, three typical flow patterns of thermal mixing in tee pipes were simulated using the ELES method, and the results were compared with the experimental data in reference [5] to prove the validity of the ELES model. Secondly, the flow characteristics of the backflow upstream with different momentum ratios were studied by carrying out 12 groups of simulations using the ELES method, and some general rules about that were revealed. Thirdly, the temperature characteristics near the wall where the backflow appears were studied. It revealed that some regions were more prone to thermal fatigue at



the intersection of the tee upstream under specified momentum ratios due to the mixing induced by the backflow. Moreover, in order to evaluate the thermal fatigue, a frequency analysis of the fluid temperature fluctuation at specified points near the wall of three typical flow patterns was conducted, and the results showed some long-period fluctuations of low frequencies upstream, which may be related to the characteristics of the backflow. The results of this paper can be helpful for the understanding of the thermal fatigue mechanism caused by the backflow, as well as the optimization and improvement of upstream thermal fatigue in tee pipes.

## 2. Turbulence Model and CFD Methodology

### 2.1. Governing Equations of LES/WMLES

The ELES method splits the domain into RANS and LES portions. The velocity field $U(x, t)$ is decomposed into a filtered component $U(x, t)$ and a subgrid-scale component $u'(x, t)$ in the LES portion as below:

$$U(x, t) = \overline{U}(x, t) + u'(x, t) \tag{1}$$

Unlike the mean (time-averaged) component $U(x, t)$ in the Reynolds decomposition, here, $U(x, t)$ represents an instantaneous field. The filtered momentum equation can then be derived once the velocity decomposition is brought into the Navier–Stokes equation [23]:

$$\frac{\partial \overline{U}_j}{\partial t} + \frac{\partial \overline{U_i U_j}}{\partial x_i} = v \frac{\partial^2 \overline{U}_j}{\partial x_i \partial x_i} - \frac{1}{\rho} \frac{\partial \overline{p}}{\partial x_j} \tag{2}$$

where $p(x, t)$ is the filtered pressure field, which contains the body force influence, and the residual stress tensor arising from the subgrid-scale motion is defined as Equation (3):

$$\tau_{ij} = \rho(\overline{U_i U_j} - \overline{U}_i \overline{U}_j) \tag{3}$$

Then,

$$\frac{\partial \overline{U}_j}{\partial t} + \frac{\partial (\overline{U}_i \overline{U}_j)}{\partial x_i} = v \frac{\partial^2 \overline{U}_j}{\partial x_i \partial x_i} - \frac{\partial \tau_{ij}}{\partial x_i} - \frac{1}{\rho} \frac{\partial \overline{p}}{\partial x_j} \tag{4}$$

Closure needs to be achieved by additional modeling of $\tau_{ij}$.

In addition, the filtered continuity equation is defined as Equation (5):

$$\overline{\left( \frac{\partial U_i}{\partial x_i} \right)} = \frac{\partial \overline{U}_i}{\partial x_i} = 0 \tag{5}$$

The energy governing equation is defined as Equation (6):

$$\frac{\partial (\rho \overline{h})}{\partial t} + \overline{U}_i \frac{\partial (\rho \overline{h})}{\partial x_i} = \frac{\partial}{\partial x_i} \left[ \left( k + \frac{\mu_t c_p}{\mathrm{Pr}_t} \right) \frac{\partial \overline{T}}{\partial x_i} \right] \tag{6}$$

where $h$ represents the filtered enthalpy, $k$ is the thermal conductivity, $c_p$ is the constant pressure-specific heat of the fluid, $Pr_t$ is a turbulent Prandtl number, and $\mu_t$ is the eddy viscosity.

Back to the closure modeling of the residual stress tensor $\tau_{ij}$, it can be computed based on the Boussinesq hypothesis, and the expression is as Equation (7) [24]:

$$\tau_{ij} = -2\mu_t \overline{S}_{ij} + \frac{1}{3} \tau_{kk} \delta_{ij} \tag{7}$$

where $\mu_t$ and $S_{ij}$ are the eddy viscosity and the filtered rate-of-strain tensor, respectively. $S_{ij}$ is defined as Equation (8):

$$\overline{S}_{ij} = \frac{1}{2} \left( \frac{\partial \overline{U}_i}{\partial x_j} + \frac{\partial \overline{U}_j}{\partial x_i} \right) \tag{8}$$

The Wall-Modeled LES (WMLES) method is adopted for the entire LES domain and reduces the stringent and *Re* number-dependent grid resolution requirements of classical wall-resolved LES [19]. The original Algebraic WMLES formulation was put forward by Shur et.al. [25], which incorporates several models, such as a modified Smagorinsky model [26], the wall-damping function of Piomelli [27], and a mixing length model. The eddy viscosity $\mu_t$ in the Shur et al. model is defined as Equation (9).

$$\mu_t = \rho \min\left[ (\kappa d_w)^2, (C_{Smag}\Delta)^2 \right] \bullet S \bullet \left\{ 1 - \exp\left[ -(y^+/25)^3 \right] \right\} \tag{9}$$

where $d_w$ is the wall distance, $\kappa = 0.41$ is the Von Karman constant, and $C_{Smag} = 0.2$. $S$ is the strain rate, and $y^+$ is the normal wall inner scaling. $\Delta$ is defined as Equation (10).

$$\Delta = \min(\max(C_w \bullet d_w; C_w \bullet h_{\max}, h_{wn}); h_{\max}) \tag{10}$$

Here, $h_{max}$ is the maximum edge length for a rectilinear hexahedral cell, $h_{wn}$ is the wall-normal grid spacing, and $C_w$ is a constant 0.15.

## 2.2. Governing Equations of SST k-ω Model

The k-omega SST model was used in the RANS portion to obtain the inlet velocity and turbulence quantities. The equations of the turbulence kinetic energy ($k$) and the specific dissipation rate ($\omega$) for the SST $k$-$\omega$ model are as follows [28,29]:

$$\frac{\partial}{\partial x_i}(\rho k u_i) = \frac{\partial}{\partial x_j}\left[ (\mu + \mu_t/\sigma_k)\partial k/\partial x_j \right] + G_k - Y_k \tag{11}$$

$$\frac{\partial}{\partial x_i}(\rho \omega u_i) = \frac{\partial}{\partial x_j}\left[ (\mu + \mu_t/\sigma_\omega)\partial \omega/\partial x_j \right] + G_\omega - Y_\omega + D_\omega \tag{12}$$

The turbulent viscosity $\mu_t$ and terms such as $G_k$, $Y_k$, $G_\omega$, $Y_\omega$, $D_\omega$ are defined as:

$$\mu_t = \frac{\rho k}{\omega}\left[ \max\left( \frac{1}{\alpha^*}, \frac{SF_2}{\alpha_1\omega} \right) \right]^{-1} \tag{13}$$

$$G_k = \min\left( \mu_t S^2, 10\rho\beta^* k\omega \right) \quad Y_k = \rho\beta^* k\omega \tag{14}$$

$$G_\omega = \frac{\alpha_\infty}{\alpha^*}\left( \frac{\alpha_0 + Re_t/2.95}{1 + Re_t/2.95} \right)\rho S^2 \quad Y_\omega = \rho\beta^*\omega^2 \tag{15}$$

$$D_\omega = 2(1 - F_1)\rho\sigma_{\omega,2}\frac{1}{\omega}\frac{\partial k}{\partial x_j}\frac{\partial \omega}{\partial x_j} \tag{16}$$

## 2.3. Physical Model

The physical problem was that two streams of water at 48 °C with *Pr* = 3.7 and 33 °C with *Pr* = 5.1 flowed into the tee pipe from different inlets and mixed. The water experiments were conducted by Kamide et.al. In the experiment, hot and cold water entered the test section from a horizontal main pipe with a diameter of 150 mm ($D_m$) and a vertical branch pipe with a diameter of 50 mm ($D_b$), respectively. The main pipe of the test section had 18 $D_m$ of the upstream length to the tee and it was 10 $D_b$ in the branch pipe. In this study, the computational domain was established with entrance lengths of 650 mm (4.3 $D_m$) and 175 mm (3.5 $D_b$) from the tee with the same diameters as the test section. The downstream length was set as 450 mm (3 $D_m$) in the computational domain to cover the test results. A schematic diagram of the computational domain is shown in Figure 1. The cross-section at 1 $D_m$ upstream of the tee of the main pipe was set as a RANS-LES interface, while the cross-sections at 0.5 $D_m$ and 1 $D_m$ downstream of the tee identified in Figure 1 were consistent with the test as monitor positions. In the main pipe, the upstream region divided by the RANS-LES interface was the RANS computational domain, while another one was the LES computational domain. Similarly, the cross-section at $-2.3 D_b$ upstream of

the tee of branch pipe was set as a RANS-LES interface. Besides, the coordinate origin was located at the intersection of pipe axes as shown in Figure 1.

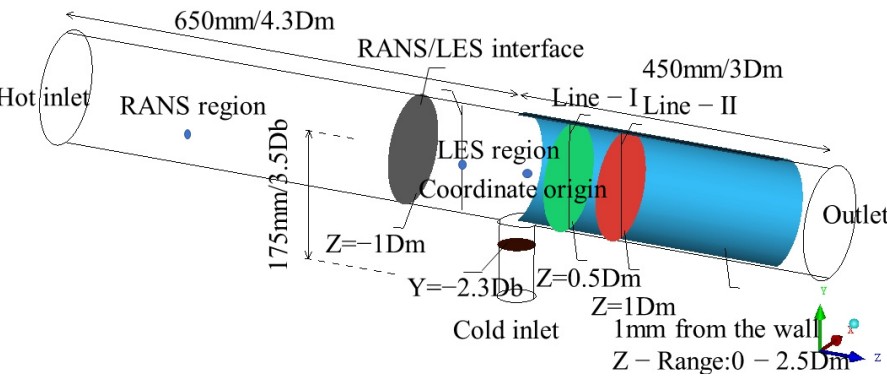

**Figure 1.** Schematic diagram of the computational domain.

### 2.4. Precursor Simulations

When the ELES simulations are carried out, unsteady fluctuations at the RANS-LES interface are required to be provided to the LES domain [19]. Therefore, precursor RANS simulations are conducted for the profiles of inlet turbulence quantities and the pre-estimation of mesh size in the central mixing zone, and the K-omega SST turbulence model is employed before carrying out the ELES. In the precursor RANS simulations, turbulent flows are conducted in pipes with a diameter of 1 $D_m$ and 1 $D_b$, respectively. In order to obtain the fully developed inlet profiles, the pipe lengths are set as 18 $D_m$ and 10 $D_b$, respectively, as the requirement stated in reference [19]. Through the precursor simulations, the profiles of inlet turbulence quantities can be obtained.

Precursor simulations of three cases with typical flow patterns are simulated in the pipes with the K-omega SST turbulence model. Details of the three cases are listed in Table 1. According to reference [5], the flow pattern can be predicted by the momentum ratio $M_R$: $M_R < 0.35$: impinging jet, $0.35 < M_R < 1.35$: deflecting jet, $M_R > 1.35$: wall jet. Furthermore, the momentum ratio $M_R$ is defined as Equation (17):

$$M_R = \frac{M_m}{M_b} = \frac{D_m D_b \rho_m V_m^2}{\pi/4 D_b^2 \rho_b V_b^2} \tag{17}$$

**Table 1.** Conditions of three typical cases.

| Case | Average Velocity (m/s) | | Momentum Ratio ($M_R$) | Flow Pattern |
|---|---|---|---|---|
| | Main ($V_m$) | Branch ($V_b$) | | |
| W-ref | 1.46 | | 8.1 | Wall jet |
| D-ref | 0.46 | 1 | 0.8 | Deflecting jet |
| I-ref | 0.23 | | 0.2 | Impinging jet |

### 2.5. Meshing Requirement

2.5.1. Meshing Requirement near the Wall in the LES Portion

Quite a number of suggestions on the meshing requirements of LES have been given in the related literature. NJ Georgiadis et al. [30] gave the usual values of the grid spacing in three directions: $\Delta z^+ = 50$–150, $\Delta x^+ = 15$–40, and $N_y = 60$–80 beginning with $\Delta y^+ \leq 1$, where $x$, $y$, and $z$ are the spanwise direction, the wall normal, and the streamwise, $\Delta i^+$ is the non-dimensional grid spacing defined as $\Delta i^+ = u_\tau \Delta i / v$ ($i$ refers to $x$, $y$, and $z$), and $N_y$ is the number of cells to cover the boundary layer in the wall-normal direction. This is a typical requirement for classic wall-resolved LES, and the values of the actual grid spacing ($\Delta i$) would be *Re*-dependent in practice.

Meanwhile, in applications with a wall-modeled LES formulation, the requirement can be relaxed. In the wall-normal direction, although a coarser $\Delta y^+$ value such as $\Delta y^+ \approx 20$ [31] can be tolerable, a near-wall resolution of $\Delta y^+ = 1$ was taken in the current work and the number of cells across the boundary layer was set as $N_y = 60$ as recommended by ANSYS. A typical wall-normal grid spacing $\Delta y$ of the main pipe was set as 0.02 mm to ensure the near-wall resolution in the current work had a *Re* number of approximately $3.9 \times 10^5$. In the other two directions, the grid spacing $\Delta x \approx \Delta z \approx (0.05\text{–}0.1)\delta$ was adopted in the current work as recommended by ANSYS, where $\delta$ (mm) is the per boundary layer thickness. The $\Delta x / \Delta z$ of the main pipe was fixed at 2.25 mm/3.75 mm in the current work, and $(\Delta x^+, \Delta z^+) \approx (113, 188)$ in wall units with a *Re* number of approximately $3.9 \times 10^5$.

### 2.5.2. Meshing Requirement in the Mixing Zone Downstream in the LES Portion

To meet the mesh resolution of the separating shear layer in the mixing zone downstream, Addad et al. [32] recommended a minimal LES mesh resolution with which the mesh size $\Delta$ was taken as max $(\lambda_R, L_t^R/10)$, where the Taylor microscale $\lambda_R$ and the turbulent energy length $L_t^R$ were both obtained from RANS estimations, as seen in Equations (18) and (19).

$$\lambda_R = \sqrt{10k\nu/\varepsilon} \tag{18}$$

$$L_t^R = \frac{k^{3/2}}{\varepsilon} = \frac{k^{1/2}}{0.09\omega} \tag{19}$$

Kuczaj et al. [33] concluded that the mesh size of $\Delta \approx \lambda/3$, where $\lambda$ was the Taylor microscale obtained from a posteriori LES analysis, can give sufficiently accurate results, and the value of $\lambda/3$ was found to be equivalent to the RANS Taylor microscale $\lambda_R$. X Fang et al. [31] recommended a mesh size of $\Delta/\eta_R < (50\text{–}100)$ for LES mixing layers, where the Kolmogorov scale $\eta_R$ can be calculated as Equation (20):

$$\eta_R = L_t^R (k^2/\varepsilon\nu)^{-3/4} \tag{20}$$

Here, when non-equilateral cells are used, the mesh scale $\Delta$ refers to max $(\Delta x, \Delta y, \Delta z)$; in addition, $k$, $\varepsilon$, and $\omega$ are the turbulent kinetic energy, the turbulent dissipation energy, and the specific dissipation rate, respectively. Eventually, the mesh scale of the free shear portion was set as Equation (21):

$$\Delta = min\ (\lambda_R, L_t^R/10,\ 50\eta_R) \tag{21}$$

As precursor RANS simulations were performed, a typical mesh size was obtained by the above estimator, that is $\Delta = 2.5$ mm in the free shear portion. Then, hexahedral structure grid elements were constructed in the current work.

### 2.5.3. Meshing Requirement in the RANS Portion

After the details of meshing requirements in the LES portion are emphasized, the resolution requirements in the RANS portion should be determined. It is recommended that the wall boundary layer should be covered with (20–30) cells with $y^+ \sim 1$, and the minimum resolution across free shear flows can follow Equation (22).

$$\Delta_{\max} \leq 0.05D \tag{22}$$

Therefore, we set $\Delta = 7.5$ mm in the RANS portion as the coarser mesh can be accepted in this domain.

Finally, a typical grid structure with a total of approximately 4 million elements is shown in Figure 2. The A–A cross-section is in the LES portion and the B–B cross-section is in the RANS portion. It can be seen that the mesh in the RANS portion is much coarser than that in the LES portion, which will reduce the number of grids in the whole computational domain, as well as the computation cost.

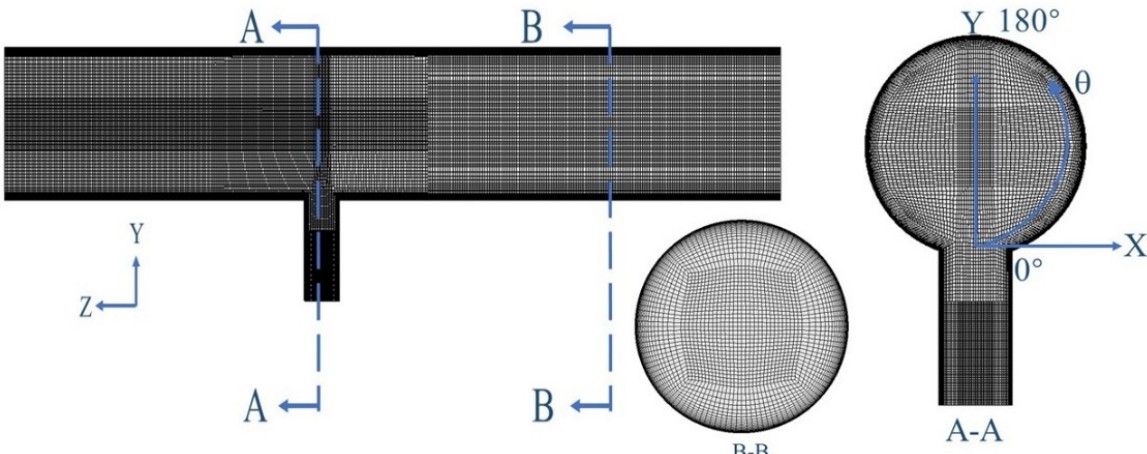

**Figure 2.** Schematic diagram of the computational mesh.

In order to assess the grid independence, four types of grids shown in Figure 3 were made according to the above criteria. They had approximately 0.5, 1.6, 4.0, and 6.4 million cells, and the information is listed in Table 2 (cases are defined as Case-A, Case-B, Case-C, and Case-D in ascending order of the grid's number). The cell thickness of the first layer from the pipe's inner surface is marked as $\Delta y_1$. Additionally, the grid independence is covered in more detail in Section 3.1.

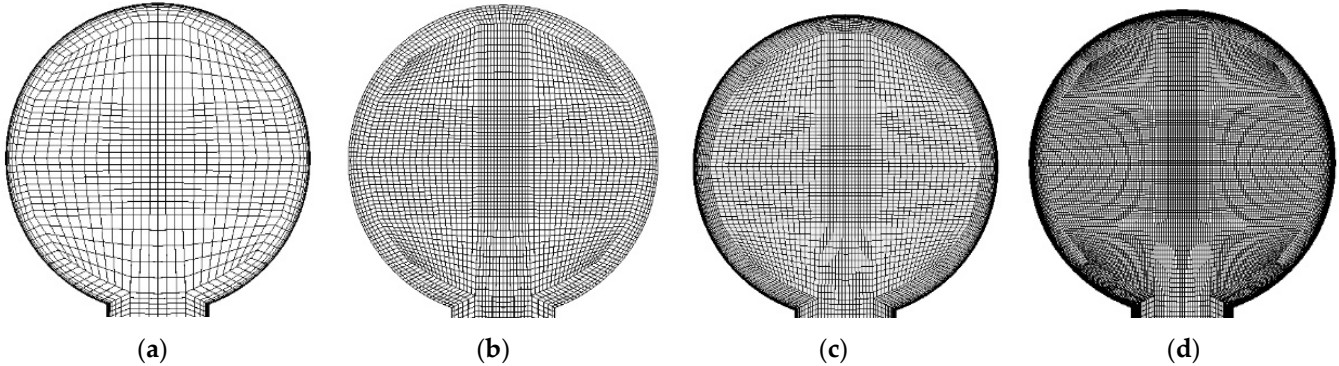

|     |     |     |     |
|:---:|:---:|:---:|:---:|
| (**a**) | (**b**) | (**c**) | (**d**) |

**Figure 3.** Computational grids on pipe cross sections in the LES portion: (**a**) Case-A with approximately 0.5 million cells; (**b**) Case-B with approximately 1.6 million cells; (**c**) Case-C with approximately 4.0 million cells; (**d**) Case-D with approximately 6.4 million cells.

**Table 2.** Information for grid independence.

| Case | Treatment near the Wall | Treatment in the Free Shear Layer | Criteria |
|:---:|:---:|:---:|:---:|
| Case-A | $N_y = 30$, $\Delta y_1 = 0.02$, $y^+ \sim 1$ | $\Delta < 5$ | More relaxed requirements put forward by NJ Georgiadis et al. [28] and Equation (21) |
| Case-B | $N_y = 60$, $\Delta y_1 = 1$, $y^+ \sim 50$ | $\Delta < 2.5$ | Strict requirements put forward by NJ Georgiadis et al. [28] and Equation (21) |

**Table 2.** *Cont.*

| Case | Treatment near the Wall | Treatment in the Free Shear Layer | Criteria |
|---|---|---|---|
| Case-C | $N_y = 60$, $\Delta y_1 = 0.02$, $y^+ \sim 1$ | $\Delta < 2.5$ | More relaxed requirements put forward by NJ Georgiadis et al. [28] and Equation (21) |
| Case-D | $N_y = 80$, $\Delta y_1 = 0.02$, $y^+ \sim 1$ | $\Delta < 1.5$ | Strict requirements put forward by NJ Georgiadis et al. [28] and Equation (21) |

*2.6. Boundary Conditions and Numerical Setting*

The vortex method [34] was adopted to generate turbulent fluctuations at RANS-LES interfaces in the simulations. The number of vortices was set to be large enough to affect all spots on the face zone. For the inlet boundaries, the profiles of velocity and turbulence quantities obtained from precursor RANS simulations were specified. For the solid boundaries, the pipe walls were set under non-slip and adiabatic conditions.

As recommended in reference [19], the time steps should be chosen to achieve a Courant number of CFL~1 in the LES portion. Therefore, the time step was set as 0.001 s in our simulations, which leads to CFL~1 in the central mixing zone. The mass conservation was most concerned in the process of convergence, and the mass residuals decreased by more than 10 orders of magnitude each time step with 20 inner iteration loops. Other numerical settings and related information in the current simulations are listed in Table 3.

**Table 3.** A summary of the current numerical simulation.

| Numerical Settings | | |
|---|---|---|
| Spatial discretization | Momentum | Bounded central differencing in LES; 2nd order upwind in RANS |
| | K & $\Omega$ | 2nd order upwind in RANS |
| | Gradients | 1st order upwind |
| | Pressure | 2nd order upwind |
| Time Discretization | Transient formulation | Bounded second order implicit |
| | Time advancement | Iterative time advancement with SIMPLEC scheme |
| | Time step interval | 0.001 (s) |
| | Iteration every 1 time-step | 20 iterations |
| Models | RANS portion | K-omega SST |
| | LES portion | WMLES |
| Precursor simulations | RANS domain | K-omega SST |

As for the physical properties of the water, the density, specific heat capacity, and coefficient of thermal conductivity change slightly with the temperature, hence, the three physical properties are treated as constant. However, the dynamic viscosity changes a great deal with the temperature and it seems that the dynamic viscosity is approximately linear with a temperature between 33 °C and 48 °C; therefore, a UDF was constructed, and the detailed code can be seen in Appendix A.

**3. Simulations with the Numerical Method ELES**

*3.1. Meshing and Independence Validations for Grid*

The different mesh requirements used in simulation may influence the computational accuracy and the computational time. Therefore, a grid independence check was performed

before the numerical computation to establish an appropriate grid structure in order to maintain a balance between computational accuracy and time. Four cases using the different mesh requirements listed in Table 2 were selected for comparison under the same computational conditions.

Additionally, two statistical variables, which were mean axial velocity $V_Z$ and axial velocity fluctuation intensity at position $Z = -0.53 \, D_m$ in the W-ref case, were taken as a comparison. Both the mean velocity and the intensity were normalized by the bulk velocity in the main pipe, $V_m$. Definitions are as below:

$$\overline{V}_z^* = \frac{\overline{V}_z}{V_m} \tag{23}$$

$$V_{z-sd}^* = \frac{1}{V_m} \sqrt{\frac{\sum_i \left(V_z^i - \overline{V}_z\right)^2}{N}} \tag{24}$$

where $N$ is the total number of sampling data and $V_z^{\,i}$ is an instantaneous value at a certain position along the vertical line.

Considering that the frequency of the particle image velocimetry (PIV) used in reference [5] was 15 Hz (0.066 s) and approximately 17 s for the measurement, just for comparison, the sampling interval was set as 0.066 s (66 time-steps) and a period of approximately 10 s was taken for statistics after calculation convergence in the current work.

The simulation results of four grids are shown in Figure 4. It shows that the results reproduce the tendency of the distribution pretty well. At $-0.53 \, D_m$ upstream of the tee, there is no big difference between the mean axial velocity among the grids. Although the results of the four cases vary to some degree, the calculated mean axial velocity is in good agreement with the measured data. However, the results of the axial velocity fluctuation intensity in Case-C and Case-D show better than that of Case-A and Case-B. Case-A underestimated the experimental data, while Case-B performed worse predictions near the wall than others. Considering the computational cost, Case-C seems the best choice among the four grids since it not only has high enough accuracy but also saves computational resources. Therefore, the grid structure of Case-C was selected in the present research.

### 3.2. Turbulence Generation and Structure

Three cases with typical flow patterns listed in Table 1 were simulated with ELES. Before processing the validation of the ELES model, some work needs to be completed such as checking the turbulence generation and structure.

By vorticity post-processing, Figure 5a shows that the resolved turbulences were well-generated from the upstream interfaces in the three reference cases. Here, the vorticity (omega) has projections in the Cartesian coordinate system as seen in Equation (25):

$$\Omega_x = \frac{\partial V_z}{\partial y} - \frac{\partial V_y}{\partial z}, \Omega_y = \frac{\partial V_x}{\partial z} - \frac{\partial V_z}{\partial x}, \Omega_z = \frac{\partial V_y}{\partial x} - \frac{\partial V_x}{\partial y} \tag{25}$$

Additionally, the turbulence structures in the current ELES simulations need to be checked. Figure 5b shows iso-surfaces of the $Q$-criterion colored with $V_z$ in three cases. Turbulence structures are depicted to be well-developed and show no damping or disruption downstream. The definition of Q is defined as Equation (26):

$$Q = \frac{1}{2}\left(\Omega^2 - S^2\right) \tag{26}$$

where $S$ is the strain rate and $\Omega$ is the vorticity, and both are absolute values. The iso-surfaces of the $Q$-criterion shown in Figure 5b are equal to 200 [s$^{-2}$]. The above analysis reveals that the current ELES simulations provide high resolution on turbulence structures in LES zones with the vortex method adopted at the upstream interface.

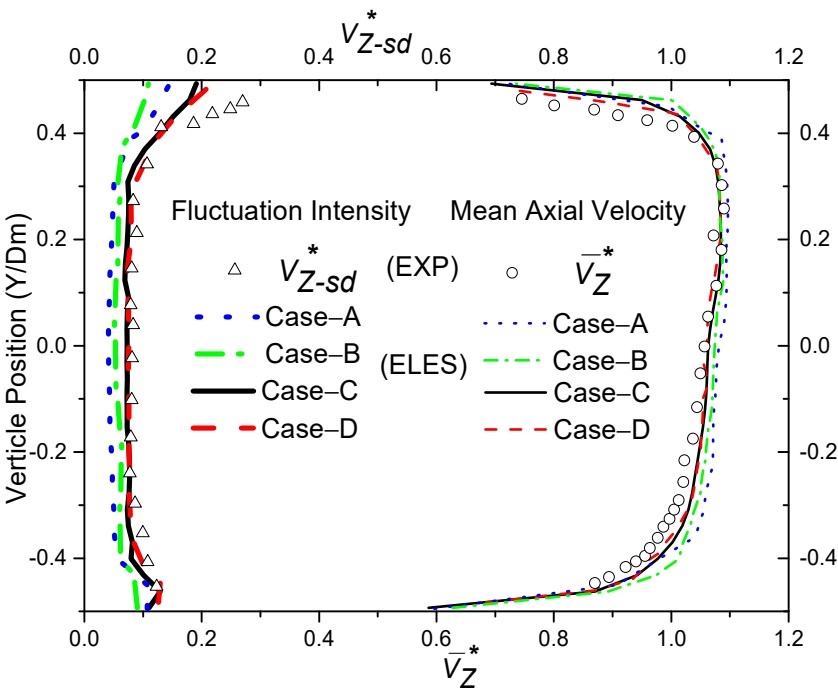

**Figure 4.** Mean axial velocity and velocity fluctuations distributions along a vertical line ($Z = -0.53\,D_m$) in the W-ref case for four types of grids.

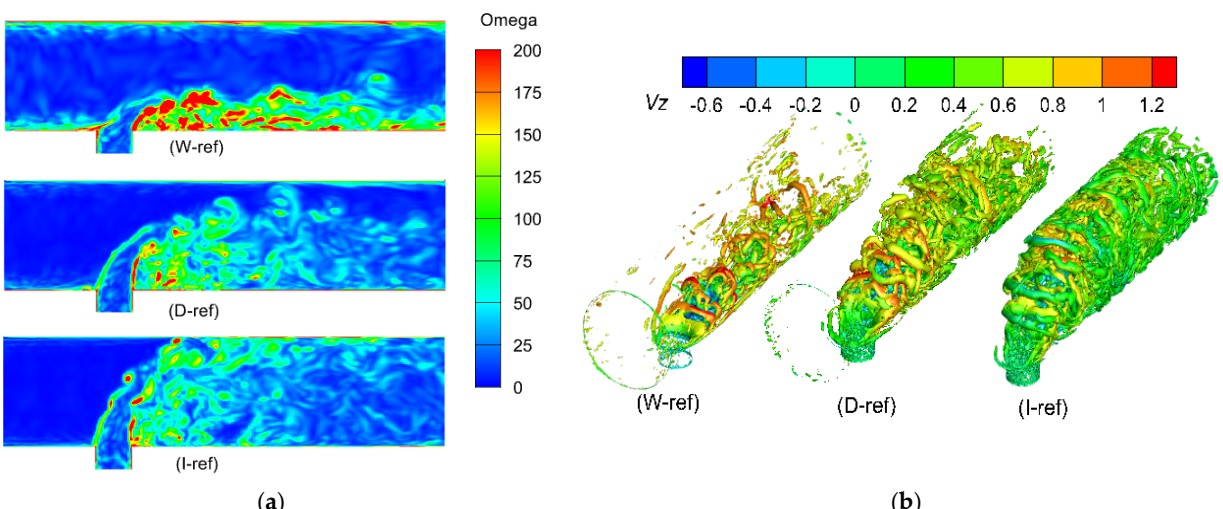

**Figure 5.** Contours of turbulence generation and structure in three cases: (**a**) Vorticity contours, (**b**) $Q$-criterion colored with $V_z$ in the cases.

### 3.3. Validation of the Numerical Method ELES

3.3.1. The Contours of Temperature and Velocity Distributions Downstream

The distributions of temperature and fluctuation intensity on a half-cylindrical surface 1 mm apart from the main pipe wall and two cross-sections downstream (as shown in Figure 1) were obtained by simulation. Figures 6–8 show comparisons of these statistical results and the experimental ones. The left sides of Figures 6–8 are the LES results of Kamde et al. [5]. using the AQUA code, which is an in-house thermal hydraulic simulation code [35]. The right sides are the ELES simulation results using FLUENT 17.0.

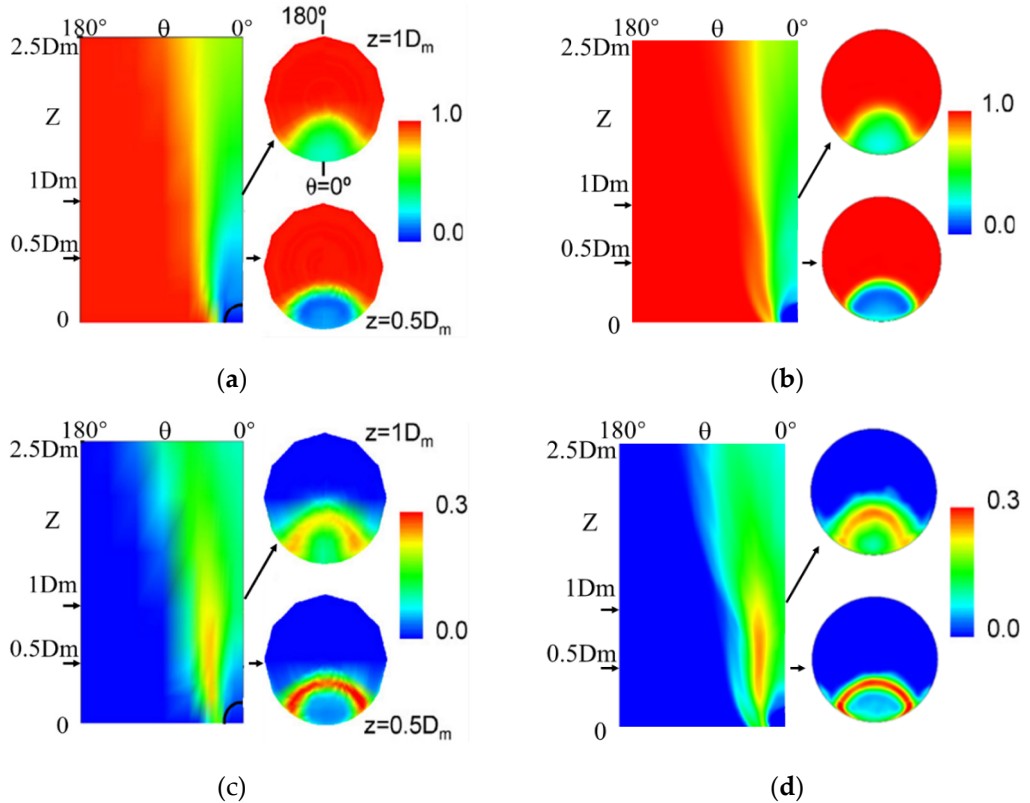

**Figure 6.** Temperature distribution downstream of W-ref case using LES and ELES: (**a**) Temperature $T^*$ distribution using LES; (**b**) Temperature $T^*$ distribution using ELES; (**c**) Temperature $T_{sd}^*$ distribution using LES; (**d**) Temperature $T_{sd}^*$ distribution using ELES.

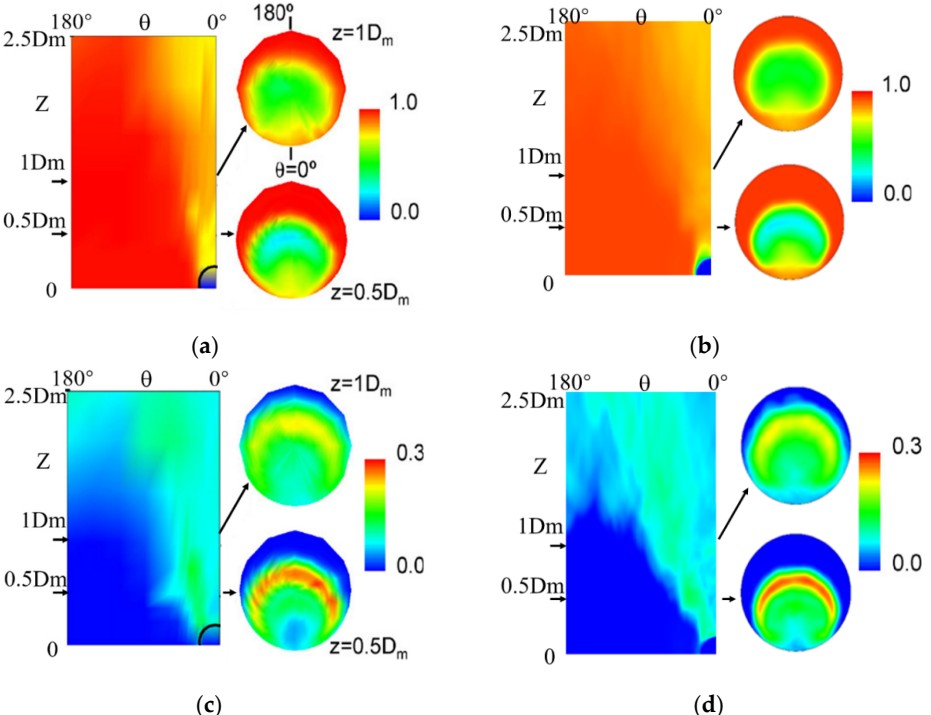

**Figure 7.** Temperature distribution downstream of D-ref case using LES and ELES: (**a**) Temperature $T^*$ distribution using LES; (**b**) Temperature $T^*$ distribution using ELES; (**c**) Temperature $T_{sd}^*$ distribution using LES; (**d**) Temperature $T_{sd}^*$ distribution using ELES.

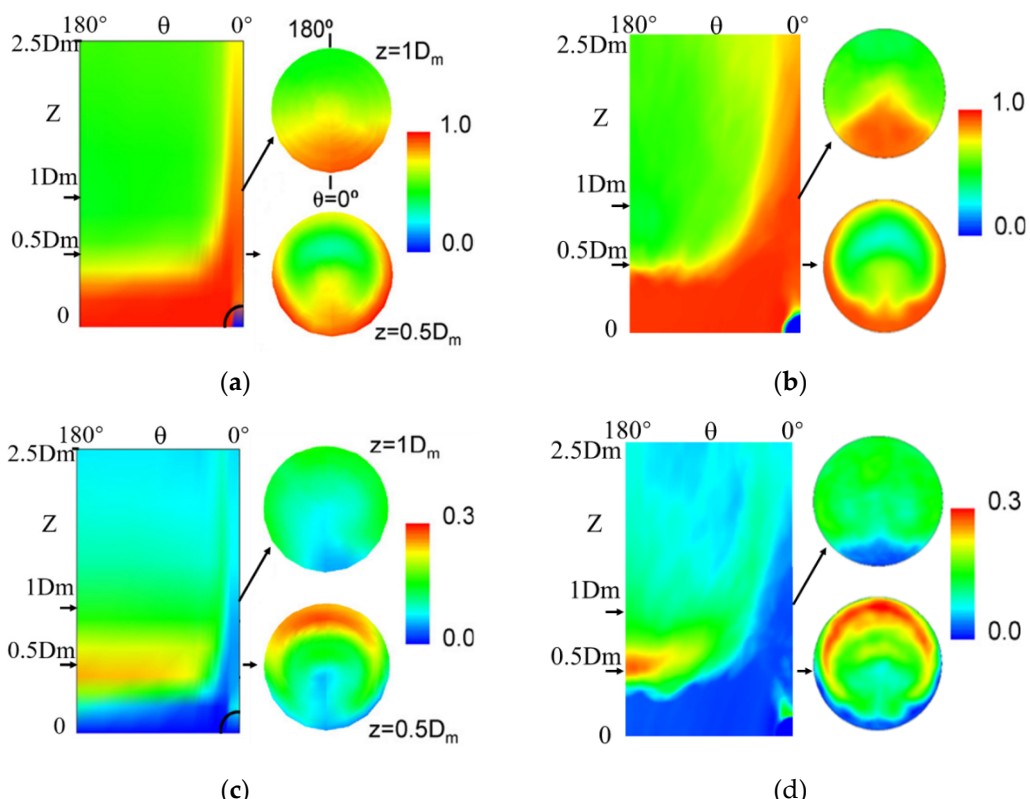

**Figure 8.** Temperature distribution downstream of I-ref case using LES and ELES: (**a**) Temperature $T^*$ distribution using LES; (**b**) Temperature $T^*$ distribution using ELES; (**c**) Temperature $T_{sd}^*$ distribution using LES; (**d**) Temperature $T_{sd}^*$ distribution using ELES.

The data displayed in Figures 6–8 were normalized with the temperature difference as seen in Equations (21) and (22):

$$\overline{T}^* = \frac{\overline{T} - T_b}{T_m - T_b} \tag{27}$$

$$T_{sd}^* = \frac{1}{T_m - T_b}\sqrt{\frac{\sum_i \left(T_i - \overline{T}\right)^2}{N}} \tag{28}$$

Here, $T_b$ is the branch temperature and $(T_m - T_b)$ is the temperature difference fixed at 15 °C, $T$ is the time-averaged temperature at a certain position on a surface, and $N$ is the total number of sampling data. Considering that the sampling frequency was 100 Hz and the measurement time period was 480 s in reference [5], just for comparison, the sampling interval was set as 0.01 s (10 time-steps), and a period of approximately 100 s was taken for statistics after calculation convergence in the current work.

It can be found that the numerical values and distributions in Figures 6–8 show good agreement. It can be clearly seen in Figures 6–8 that the thermal stripping phenomenon appears and the high fluctuation intensity occurs around the cold fluid in contact with the hot. In the W-ref case, the high-intensity region touched the pipe wall to form a half-circle big-value area at each radial cross-section, while in the deflecting jet case, the high-intensity region was separate from the pipe wall because the cold fluid from the branch pipe flowed into the central part of the main pipe. On the other hand, the cold jet from the branch pipe reached the opposite side pipe wall in the impinging jet case, and the high-intensity region then appeared in the upper half of the main pipe and decreased rapidly in the axial direction.

As can be seen from the distribution results in radial cross-sections, along the main flow direction downstream in all three cases, the temperature range narrowed and the

maximum fluctuation intensity decreased, which suggested gradually sufficient mixing was obtained with the flow.

The main difference between the ELES results and the LES ones conducted by Kamide et al. [5] shown in Figures 6–8 is that the temperature distributions near the wall of ELES are much clearer than that of LES, which indicates that ELES is better able to capture temperature fluctuations near the wall.

### 3.3.2. Temperature and Velocity Distributions Downstream along Vertical Lines

Spatial distributions of the axial velocity component $V_z$, the temperature, and their fluctuation intensities along the vertical lines at position $Z = 0.5$ and $1\ D_m$ in the W-ref case were compared and are shown in Figure 9 (vertical lines named 'line-I' and 'line-II' are shown in Figure 1).

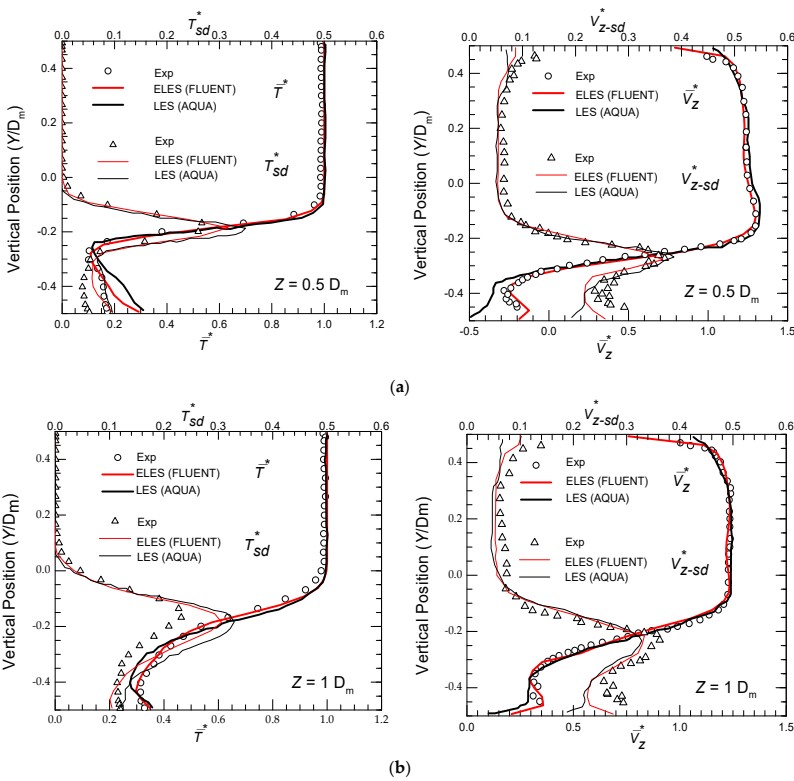

**Figure 9.** Temperature and velocity distributions along vertical lines in the W-ref case: (**a**) Line−I, = 0.5 $D_m$; (**b**) Line−II, Z = 1 $D_m$

The calculated distributions were in good agreement with those in the experiment. The fluctuation intensities were well-simulated to obtain peak values at positions where the physical quantity had a sudden change along the vertical direction. The distribution trends of temperature were similar to that of the axial velocity component, and both their obvious features appeared at the position around $Y = −0.2\ D_m$. The velocity fluctuation intensities were slightly underestimated within ELES simulations, and it was considered that the fluctuations were actually enhanced in the experiments due to the dense arrangement of quite a number of thermocouples.

In addition, the peak values of the temperature fluctuation intensity were overestimated in ELES analyses, especially at the downstream position $Z = 1\ D_m$ (the same in AQUA analyses performed by Kamide et al. [5]), which indicated that it became thermal mixing in the actual flow likely because of the enhancement of the velocity field disturbance downstream during the experiment. In addition, it obtained more appropriate results near the wall with ELES, especially for the capture of the velocity fluctuation intensity, than that in AQUA analyses performed by Kamide et al. [5].

## 4. Results and Discussion

### 4.1. The Phenomena of Backflow

As a comparison to the LES results conducted by Kamide et al. [5] shown in Figure 10a, mean axial and vertical velocities were used to depict the time-averaged streamlines in the y–z plane in the W-ref case, and the characteristics of the jet bending and eddies' formation obtained with ELES are shown in Figure 10b, which show good agreement. On the other hand, due to the prediction ability near the wall of ELES, it was found that backflow happened near the wall around the tee region, which had not been mentioned in the LES results conducted by Kamde et al. [5]. It was noteworthy that these phenomena were not special cases at certain moments as they were mean results. It can be seen as a vortex generated by backflow, which drives the mixing of hot and cold flows.

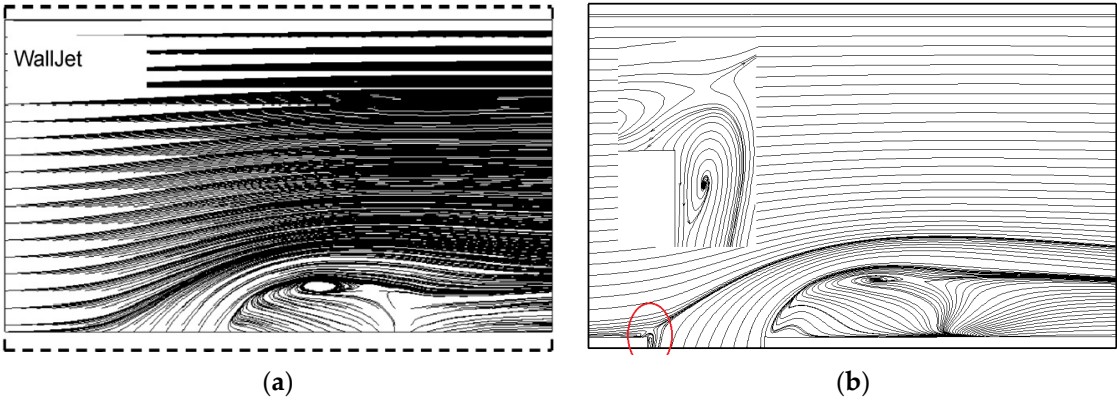

(**a**)       (**b**)

**Figure 10.** Time-averaged streamlines in y-z plane of W-ref case at X = 0. (**a**) LES results conducted by Kamde et al. [5]; (**b**) ELES results.

In order to study the influence of temperature fluctuation caused by the backflow near the wall around the tee region, a y–z plane of W-ref case at $X = 0$ was taken to form the contour of temperature fluctuation intensity, the result of which is shown in Figure 11. Figure 11a shows the overview of temperature fluctuations around the right-angle structure upstream of the tee, and temperature fluctuations in the region approximately 1 mm away from the wall are shown in Figure 11b. It can be found from Figure 11 that the temperature fluctuation intensity is close to 0.3, which is almost the same downstream of the pipe shown in Figures 6–8. Obviously, the right-angle structure upstream where the welding exists is exposed to temperature fluctuations, which makes the thermal stress-sensitive area more prone to fatigue failure. Therefore, the phenomena of the backflow in the tee region are vital to the study of thermal fatigue and need more attention.

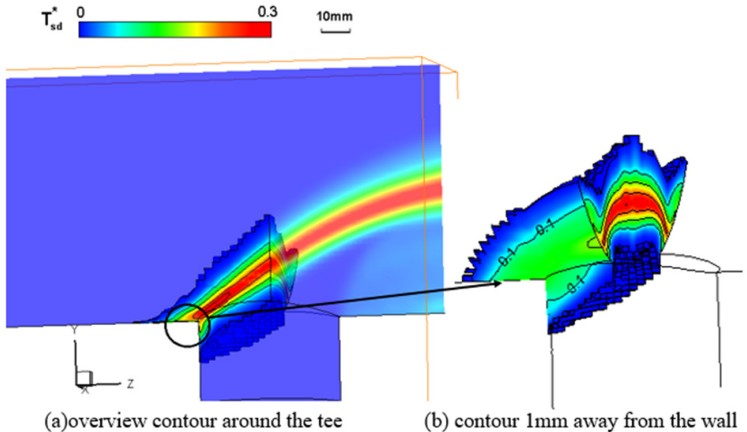

(a)overview contour around the tee       (b) contour 1mm away from the wall

**Figure 11.** Temperature fluctuation intensity distribution in the plane of $X = 0$ of the W-ref case.

In the following, more detailed quantitative analyses are further carried out to form a comprehensive understanding of the backflow in thermal mixing and the relationship between fatigue and backflow in the tee.

### 4.2. Characteristics of Backflow in the Tee

Backflow is widespread in non-isothermal mixing in the tee, and these phenomena are closely related to the momentum ratio of the main to branch flow. Therefore, to fully study these phenomena and understand the mechanism of the backflow in the tee, 12 groups of simulations with different momentum ratios were carried out and the flow field characteristics were analyzed. Detailed information on the simulations is listed in Table 4.

**Table 4.** Groups of simulations with different momentum ratios with average branch velocity $V_b = 1$ m/s.

| Case | Average Main Velocity ($V_m$) (m/s) | Momentum Ratio ($M_R$) | Flow Pattern |
| --- | --- | --- | --- |
| Case1 | 0.11 | 0.05 | Impinging jet |
| Case2 | 0.16 | 0.1 | Impinging jet |
| Case3 | 0.20 | 0.15 | Impinging jet |
| Case4 | 0.23 | 0.2 | Impinging jet |
| Case5 | 0.46 | 0.8 | Deflecting jet |
| Case6 | 1.46 | 8.1 | Wall jet |
| Case7 | 1.74 | 11.5 | Wall jet |
| Case8 | 1.76 | 11.8 | Wall jet |
| Case9 | 1.78 | 12 | Wall jet |
| Case10 | 1.85 | 13 | Wall jet |
| Case11 | 1.95 | 14.5 | Wall jet |
| Case12 | 2.9 | 32 | Wall jet |

The streamline diagrams of the three impinging jet cases were compared and the results are shown in Figure 12a–c. The streamline diagrams were drawn in the y–z plane at $X = 0$. Figure 12a–c shows that when the flow rate of branch flow is high enough, the upper fluid of the main flow is squeezed into a vortex and the phenomena of the backflow appear near the opposite sidewall. The higher the flow rate, the larger the eddy and the more obvious the phenomena of the backflow. When $M_R \leq 0.1$, the backflow with large eddies occurs near the opposite sidewall upstream. The results are similar to the experiments conducted by several researchers, e.g., Mei-Shiue Chen et al. [36], C.H. Lin et al. [37], G.Y. Chuang and Y.M. Ferng [38,39], and Wu et al. [40] In their research, the phenomena of the backflow occurred when $M_R \leq 0.05$, $M_R \leq 0.062$, $M_R \leq 0.06$, and $M_R \leq 0.05$, respectively. The difference may be the limit of observation methods in the experiment regarding the inadequate sensitivity of temperature sensors.

As the velocity of the branch flow decreases, so does the vortex, and when $M_R = 0.2$, the phenomena of the backflow that appear near the opposite sidewall almost disappeared and gradually transitioned to the place of the right-angle structure upstream of the main flow, as shown in Figure 12d. As the velocity of the branch flow decreases and that of the main flow increases, the vortex formed by backflow becomes larger at the right-angle structure upstream of the branch flow, as shown in Figure 12e–f. Compared with the size of the eddies near the opposite sidewall upstream when $M_R \leq 0.1$, the eddies shown in Figure 12d–f are relatively small, which may be the main reason that researchers have rarely observed the phenomena.

### 4.3. Temperature Characteristics near the Wall Where Backflow Appears

It can be found from the above research that different flow patterns with different momentum ratios have different characteristics of the backflow. In order to study the temperature distributions related to the backflow, detailed studies have been carried out under typical conditions in each flow pattern. From Figure 12, three typical cases with the

most obvious characteristics of the backflow were chosen as the representative of each flow pattern, which were Case1 with $M_R$ = 0.05, Case7 with $M_R$ = 0.8, and Case12 with $M_R$ = 32. Additionally, the temperature was recorded at every time step with a sampling frequency of 1000 Hz for a duration of 74.8 s.

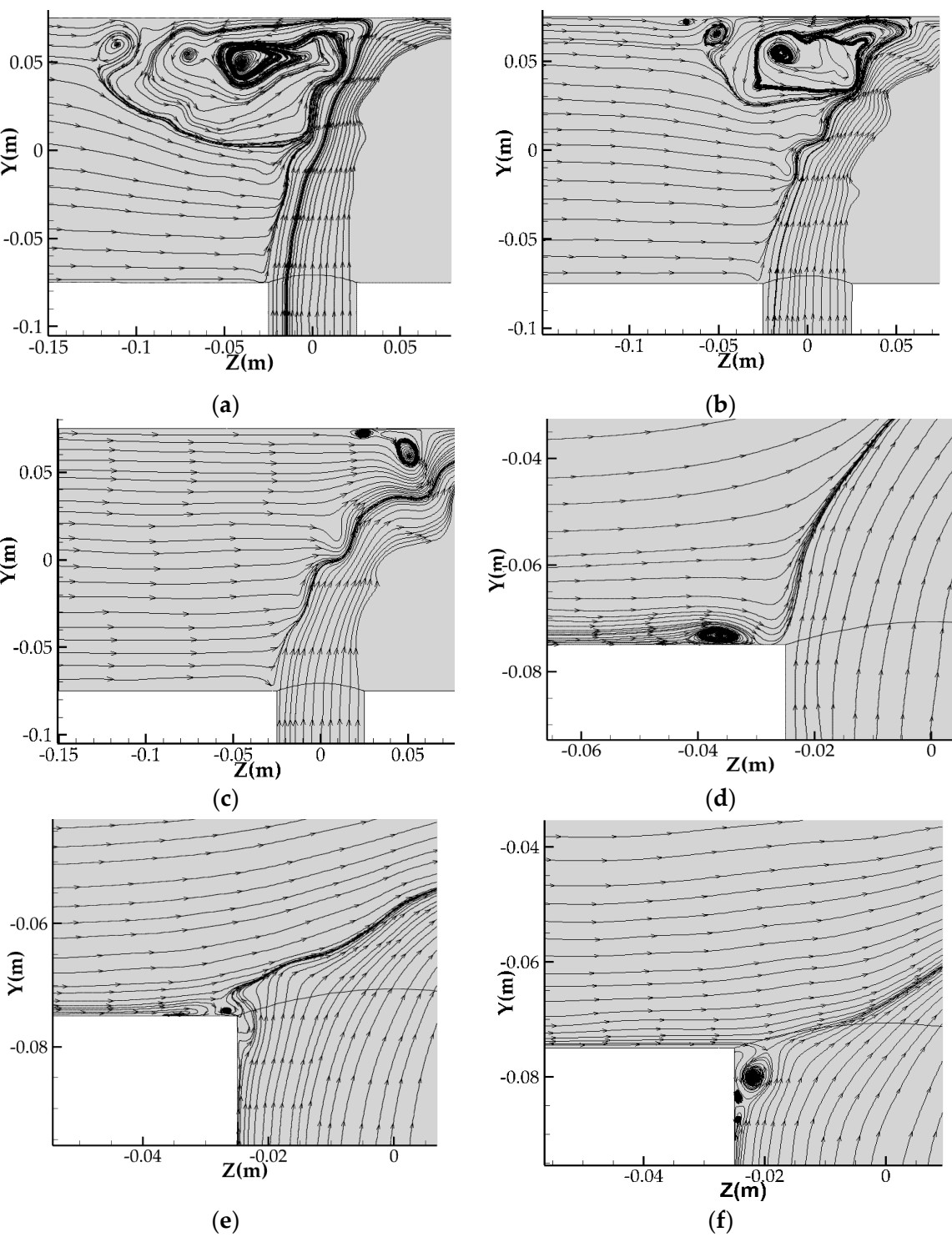

**Figure 12.** Streamlines in y–z plane of three impinging jet cases, one deflecting jet case, and tow wall jet cases at *X* = 0: (**a**) Case1 with $M_R$ = 0.05; (**b**) Case2 with $M_R$ = 0.1; (**c**) Case4 with $M_R$ = 0.2; (**d**) Case5 with $M_R$ = 0.8; (**e**) Case7 with $M_R$ = 11.5; (**f**) Case12 with $M_R$ = 32.

Axial distributions of the temperature fluctuation intensity of the above cases along corresponding lines are depicted in Figure 13. The corresponding lines of the three cases described in Figure 13 are 1 mm away from the wall. From the distribution of temperature fluctuation along line1 in Figure 13, we can see that the axial range affected by thermal striping is very wide, and it also seems that there exists more than one peak, with a maximum of approximately at $Z = -0.5 \, D_m$ upstream, which indicates that the thermal mixing caused by the backflow is complex. In addition, the temperature fluctuation intensity at $Z = -0.5 \, D_m$ shown in Figure 13 is close to that of $0.5 \, D_m$ downstream plotted in Figures 6–8. In order to further study the mixing behavior, two contours were plotted at the cross-sections of $Z = -0.5$ and $-0.2 \, D_m$ upstream as shown in Figure 14a,b. As we can see from Figure 14a,b, thermal striping appears near the opposite wall of the branch pipe, and the temperature fluctuation intensities in some locations are relatively high such as points near the wall at $105°$ and $180°$ of the cross-section at $Z = -0.2 \, D_m$ and the point near the wall at $180°$ of the cross-section at $Z = -0.5 \, D_m$.

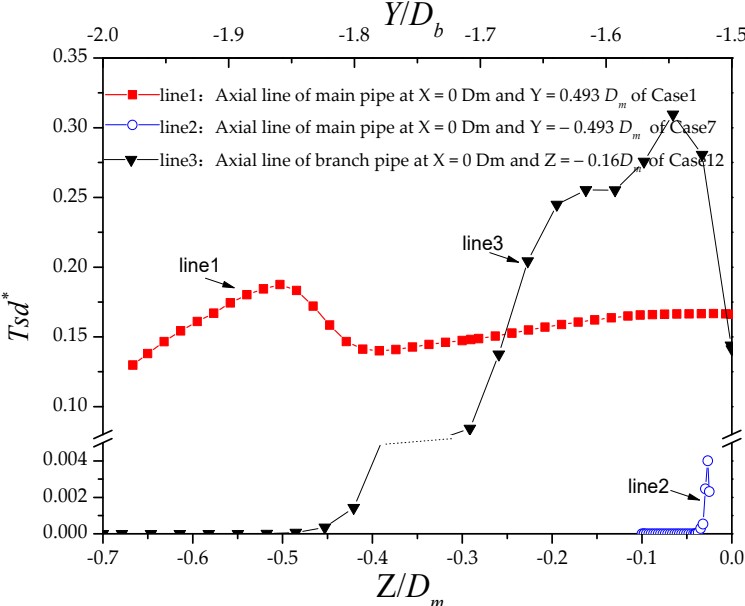

**Figure 13.** Axial distribution of temperature fluctuation intensity along corresponding lines.

From the distribution of temperature fluctuation along line2 in Figure 13, we can see that the temperature fluctuation intensity is relatively low and the axial range affected by thermal striping is narrow, which indicates that the thermal mixing is not fully developed although there exists a relatively larger eddy as shown in Figure 12d. The main reason for this phenomenon may be that the momentums of the main and branch flows are similar as $M_R = 0.8$. When each flow blocks the other one in the tee with $M_R \sim 1$, there is not enough power for one to penetrate upstream of the other one. A contour plotted at the cross-section of $Z = -0.18 \, D_m$ upstream as shown in Figure 14c reveals the above view. Therefore, the depth of penetration of one to another is dependent on the momentum ratio.

From the distribution of temperature fluctuation along line3 in Figure 13, we can see that although the axial range affected by the thermal striping is narrow, the temperature fluctuation intensity is relatively higher compared with that of Case1 and Case7. With the penetration depth, the temperature fluctuation intensity tends to increase and then decrease, and a peak of approximately 0.32 appears at the location of $Y = -1.55 \, D_b$. Figure 14d shows the temperature distributions at $Y = -1.55 \, D_b$. It can be seen that the phenomenon of thermal striping is very significant, and the radial range affected by the thermal striping is wide as it is from $-90°$ to $90°$. Meanwhile, the fluctuation peak appears at $\theta = 45°$ in Figure 14d.

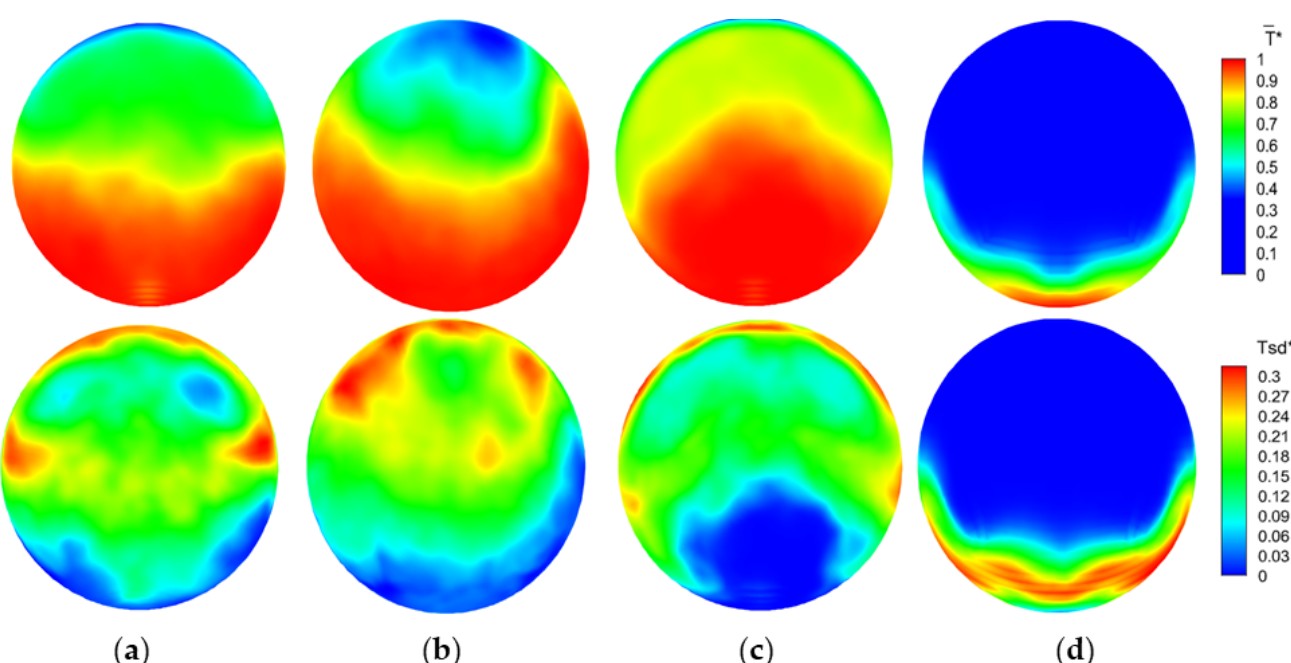

**Figure 14.** Temperature distribution upstream (the upper figures show temperature $T^*$ distribution and the lower figures show temperature $T_{sd}^*$ distribution): (**a**) A cross-section at $Z = -0.2\ D_m$ in case1; (**b**) a cross-section at $Z = -0.5\ D_m$ in case1; (**c**) a cross-section at $Z = -0.18\ D_m$ in case7; (**d**) a cross-section at $Z = -1.55\ D_b$ in case12.

In order to analyze the frequency characteristics (PSD), the FFT method was applied to the data extracted over a period of 74.8 s and presented in Figure 15. The prominent frequency components were observed in all analyses in Figure 15. At $Z = -0.5\ D_m$, $\theta = 180°$ of Case 1 in Figure 15, frequency peaks appeared at 0.5 Hz, 0.8 Hz, 1.2 Hz, 1.97 Hz, 2.53 Hz, 3.3 Hz, 6.31 Hz, etc.; at $Z = -0.2\ D_m$, $\theta = 105°$, frequency peaks appeared at 0.47 Hz, 1.26 Hz, 2.63 Hz, 5.68 Hz, etc.; while at $Z = -0.2\ D_m$, $\theta = 180°$, frequency peaks appeared at 0.48 Hz, 1.27 Hz, 2.60 Hz, 5.78 Hz, etc. Although the peaks did not exactly match, they were similar in the three locations. Besides, it is generally known that the frequency approaching 6 Hz in Figure 15 corresponds to the frequency of the Karman vortex street. In addition, the characteristic frequency was not very significant, at approximately 6 Hz downstream shown in references [5,10], but long-period temperature fluctuations of frequencies lower than 6 Hz upstream from the tee were found, which had also been observed by Utanohara et al. [10] downstream. As described in Smith et al. [41] and Thomas [42], long-period temperature fluctuations of a low frequency (0.1–10 Hz) would cause the most harmful thermal loads. Hence, the conditions may require more attention.

At $Z = -0.18\ D_m$, $\theta = 0°$ of Case 7 in Figure 15, frequency peaks appeared at 0.49 Hz, 0.89 Hz, 1.47 Hz, 2.75 Hz, 6.0 Hz, etc., while at $Y = -1.5\ D_b$, $\theta = 45°$ of Case 12, frequency peaks appeared at 0.95 Hz, 1.55 Hz, 3.2 Hz, 5.76 Hz, etc. The characteristic frequency of approximately 6 Hz in the two cases seems more obvious than that in case1, which is because the phenomena that occurred at the points in case7 and case12 are more similar to the phenomena of the Karman vortex street. Moreover, a long-period temperature fluctuation of a frequency lower than 6 Hz upstream (e.g., approximately 0.5 Hz, 1.2 Hz, 3.0 Hz, etc.) from the tee was observed, too. This may be related to the periodic variation of the backflow around the attached wall. The long-period temperature fluctuation of a frequency lower than 6 Hz downstream from the T-junction was also seen in the experimental investigation by Miyoshi et al. [43] and the numerical study by Utanohara et al. [10]. However, it requires further study to validate the long-period fluctuation of a frequency lower than 6 Hz upstream through more experiments.

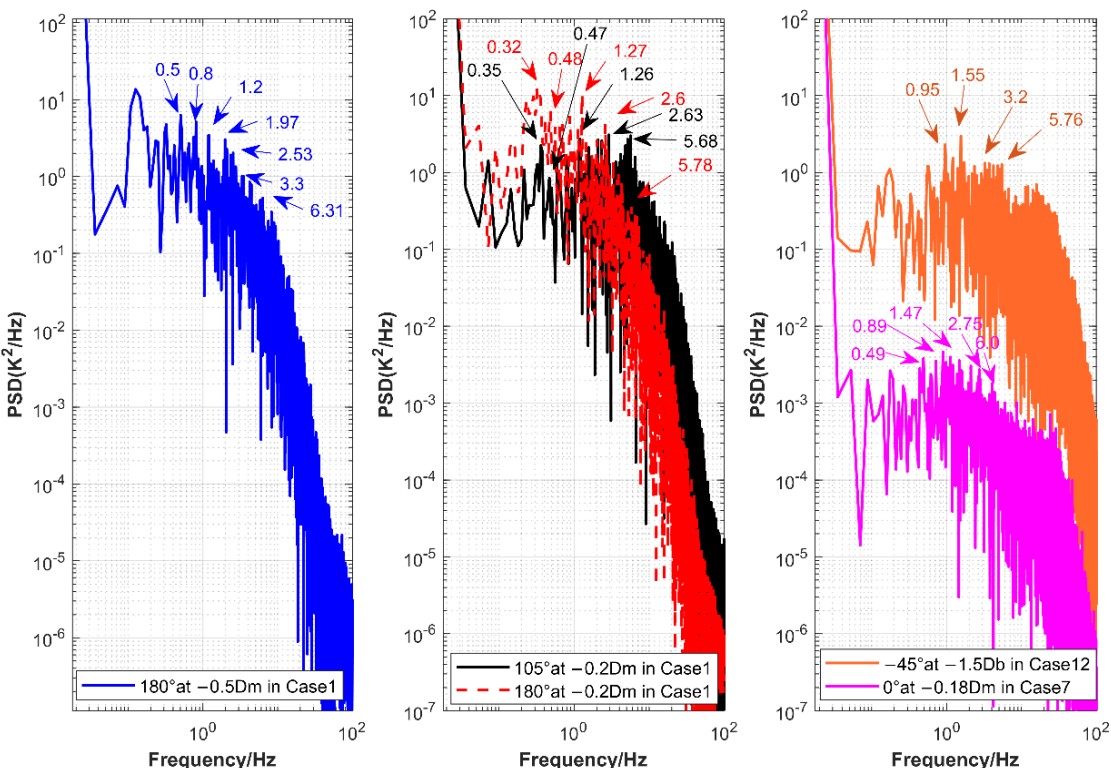

**Figure 15.** Power spectrum density of the temperature in Case1, Case7, and Case12.

In sum, by analyzing the temperature distribution of three different flow patterns with different $M_R$ values, three more regions are found to be prone to thermal fatigue at the intersection of the tee upstream. The effects of fatigue are different in the three regions. It seems that the impinging jet with a smaller $M_R$ and the wall jet with a larger $M_R$ have a greater possibility of fatigue crack on the wall around the intersection, while the effect of fatigue caused by the deflecting jet on the wall seems less significant.

## 5. Conclusions

In this paper, the mixing flow of non-isothermal fluids in tee pipes upstream is numerically studied in detail. Firstly, three typical flow patterns of thermal mixing in tee pipes (wall jet with $M_R = 8.1$, deflecting jet with $M_R = 0.8$, and impinging jet with $M_R = 0.2$) were simulated using the ELES method, and the results have been compared with the experimental data in reference [5] to prove the validity of the model. Secondly, the flow characteristics of the backflow upstream with different momentum ratios were studied by carrying out 12 groups of simulations using ELES, as well as the temperature characteristics near the wall where the backflow appears. Moreover, in order to evaluate the thermal fatigue, the frequency analysis at specified points near the wall of three typical flow patterns was estimated. The conclusions are summarized below:

(1) ELES is a valid tee pipe numerical turbulent-solving method and aptly predicts the flow patterns of non-isothermal mixing in tee pipes.

(2) The characteristics of the backflow in the tee are obviously different with different momentum ratios $M_R$. The relatively general rules are that the eddies caused by the backflow appear near the opposite sidewall of the branch pipe for cases of the impinging jet when $M_R \leq 0.1$, the eddies appear near the wall at the right-angle structure upstream of the main pipe for cases of the deflecting jet when $M_R \sim 0.8$, while the eddies appear near the wall at the right-angle structure upstream of the branch pipe for cases of the wall jet when $M_R \geq 11.5$. In addition, the size of the eddies caused by the backflow is dependent on $M_R$.

(3) The characteristics of temperature distribution and the frequency of temperature fluctuations are different under different backflow patterns with different momentum ratios $M_R$. Besides the pipe wall downstream, three more regions are found to be prone to thermal fatigue at the intersection of the tee upstream. The effects of fatigue are different in the three regions. It seems that an impinging jet with smaller $M_R$ and a wall jet with larger $M_R$ have a greater possibility of fatigue cracks on the wall around the intersection, while the effect of fatigue caused by the deflecting jet on the wall seems less significant.

(4) Long-period temperature fluctuations of frequencies lower than 6 Hz upstream (e.g., approximately 0.5 Hz, 1.2 Hz, 3.0 Hz, etc.) from the tee were observed, and they may be related to the periodic variation of the backflow around the attached wall. Besides, this requires further study to validate the long-period fluctuation of a frequency lower than 3 Hz upstream through further experiments.

**Author Contributions:** Conceptualization, H.W.; methodology, C.S.; software, Z.Z..; validation, Y.J. and F.Z.; investigation, C.S. and Z.Z.; writing—original draft preparation, C.S.; writing—review and editing, C.S..; visualization, H.W. All authors have read and agreed to the published version of the manuscript.

**Funding:** This research received no external funding.

**Institutional Review Board Statement:** Not applicable.

**Informed Consent Statement:** Not applicable.

**Data Availability Statement:** The data that support the findings of this study are available from the corresponding author upon reasonable request.

**Conflicts of Interest:** The authors declare no conflict of interest.

## Nomenclature

*Latin symbols*

| | |
|---|---|
| *cp* | specific heat at constant pressure [J/kg·K] |
| *CFL* | Courant number |
| *d* | distance [m] |
| *h* | enthalpy [J/kg] |
| *k* | turbulence kinetic energy [m$^2$/s$^2$] |
| $M_R$ | momentum ratio |
| $N_y$ | number of cells |
| $L_t{}^R$ | turbulent energy length |
| *P* | pressure [Pa] |
| *Pr* | Prandtl number |
| *Re* | Reynolds number |
| *S* | rate-of-strain tensor |
| *T* | temperature [°C] |
| $T_{sd}{}^*\ T^*$ | dimensionless temperature |
| $V_m, V_b, V_Z$ | mean axial velocity [m/s] |
| $V_{sd}{}^*\ V^*$ | dimensionless velocity |
| X,Y,Z | distance in Cartesian coordinates [m] |
| $G_k, Y_k, G_\omega, Y_\omega, D_\omega$ | constants of turbulence models |
| $y^+$ | dimensionless distance from wall |

*Greek symbol*

| | |
|---|---|
| $\varepsilon$ | turbulent dissipation energy [m$^2$/s$^3$] |
| $\mu$ | eddy viscosity [kg/m·s] |

| | |
|---|---|
| $\tau$ | shear stress [N/m$^2$] |
| $\eta_R$ | Kolmogorov scale |
| $\lambda_R$ | Taylor microscale |
| $\omega$ | specific dissipation rate [1/s] |
| $\theta$ | circumferential direction in cylindrical coordinates |
| *Subscripts* | |
| *b* | branch |
| *m* | main |
| *t* | turbulent quantity |
| *w* | wall |

## Appendix A

DEFINE_PROPERTY(user_vis, cell, thread)

```
{
    float temp, mu;
    temp = C_T(cell, thread);
    mu = 0.0045-0.000012298*temp;
    return mu;
}
```

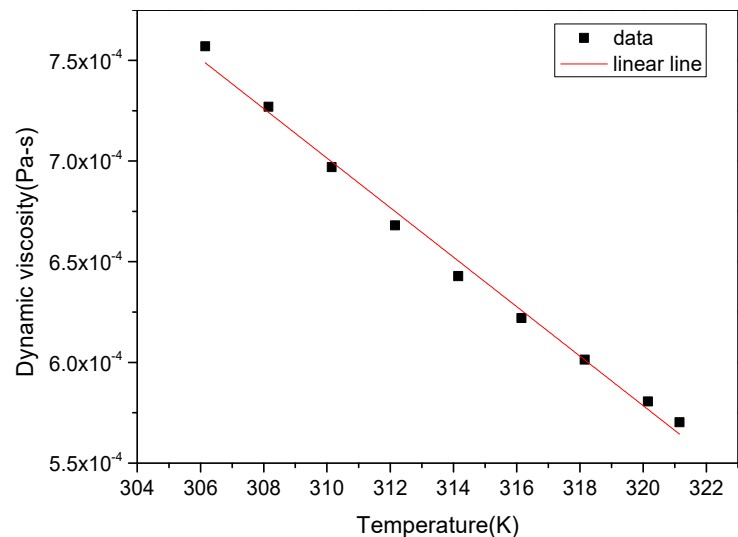

**Figure A1.** The relationship between dynamic viscosity and temperature.

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
