# Peer review of "Numerical Simulation of Non-Isothermal Mixing Flow Characteristics with ELES Method"

_applsci, doi:10.3390/app12157381_

Round 1

Reviewer 1 Report

The authors present a study examining the application of an embedded LES model to the problem of thermal fatigue and thermal striping in a mixing tee. The authors find that the embedded LES model approximates the phenomena in comparison to a full LES.

Was the flow forced or mixed convection? If mixed convection why did you select the k-omega SST model rather than an anisotropic turbulence model? 

How did you treat the physical properties of the water at the hot and cold inlet? What was the temperature difference between the hot and cold inlets? How did you approximate the influence of the temperature on the fluid motion in the Navier-Stokes equations? Density seems to be constant that there is no Boussinesq term in the Navier-Stokes equation.

What was the Prandtl number of the fluid?

How did you treat the thermal boundary layer with the embedded LES?

In Table 2 y+ values were reported, what were the T+ values, especially as T+ =f(y+). This has a strong bearing on the thermal boundary layer, as water is a high Prandtl number fluid at standard pressure and temperature, the gradient of thermal boundary layer is much steeper than the viscous boundary layer. 

A reference that may be of interest to the authors:

B.L. Smith, J.H. Mahaffy, K. Angele & J. Westin, Report of the OECD/NEA-Vattenfall T-Junction Benchmark exercise. Nuclear Energy Agency, Report No. NEA/CSNI/R(2011)5 1–92 (2011).

Suggestions for the text:

Page 3:

sufficient -> sufficiently

founded -> found

Page 5:

Cross-section -> The cross-section

Similarly, Cross-section -> Similarly, the cross-section

Page 8:

As Precursor -> As precursor

Page 10:

Vortex method -> The vortex method

Page 11:

Different mesh requirement -> The different mesh requirements

Page 19:

furtherly -> further

Page 20:

when MR  < 0.1 -> When MR < 0.1

Page 21:

rare researchers have observe the phenomena -> researchers have rarely observed the phenomena.

Page 22:

detailed researches -> detailed studies

Page 23:

monitored every time step and then -> recorded at every time step with a 

In order to furtherly study -> In order to further study

is much close -> are similar

When each flow blocks another one in the tee with MR ~ 1, there is no enough power for one to penetrate upstream of another one. -> When each flow blocks the other one in the tee with MR ~ 1, there is not enough power for one to penetrate upstream of the other one.

Page 25:

employed for 74.8 thousand data extracted during 74.8 s and Figures 21 was made. -> applied to the data extracted over the period of 74.8 s and presented in Figure 21.

Page 26:

remains further study to validify -> requires further study to validate/verify/identify  (also applies to the same statement in the conclusions)

caused by deflecting jet to the wall seems less. -> caused by deflecting jet to the wall seems less ?. ?= important/significant

or replace less with to be small

Reviewer 2 Report

This paper compares with experimental data, the applicability of the ELES model to turbulent thermal mixing was evaluated and the validity was proved. In addition, the flow characteristics of backflow upstream with different momentum ratios were studied using the ELES method as well as the temperature characteristics near the wall where backflow appears. This paper can be accepted after several major revisions:

The contributions of this work have not clearly been mentioned in the abstract

In the abstract, the authors used more general information, and more than one sentence from the abstract is repeated in the body of the manuscript, please remove this information or try to address them in a different way.

The introduction does not state clearly the novelty of the results emphasized in the manuscript.

Novelty of the work needs to be strengthened

The manuscript's introduction needs to be rephrased to be more coherent.

 This paper has some possibility of plagiarism from iThenticate. Above accumulative 30% of plagiarism, possibilities are found. Please fix this situation.

Please add the related turbulence references related to the k-ω SST (Shear Stress Transport) and RANS models

1.        Moon, J.H.; Lee, S.; Lee, J.; Lee, S.H. Numerical study on subcooled water jet impingement cooling on superheated surfaces. Case Stud Therm Eng 2022, 32, 101883, doi:10.1016/j.csite.2022.101883.

2.        Wang, J.; Gong, J.; Kang, X.; Zhao, C.; Hooman, K. Assessment of RANS turbulence models on predicting supercritical heat transfer in highly buoyant horizontal flows. Case Stud Therm Eng 2022, 34, 102057, doi:https://doi.org/10.1016/j.csite.2022.102057.

There are writing mistakes that have been observed in many places in the manuscript. 

Grammatically and typing errors are found in more than one place, so the manuscript needs to be improved.

• There is no nomenclature.

• Symbols in Figures should be presented in italic.

• The format of MDPI should be made. Only a draft without making any MDPI formats is not acceptable.

• Figures have very poor resolutions. The authors should check again for the resolution of each figure, especially for Fig. 21.

• This work has too many figures, needing to be summarized up to Fig. 15.

Round 2

Reviewer 2 Report

The reviewer’s comments are properly reflected